# Characterization of Unripe and Mature Avocado Seed Oil in Different Proportions as Phase Change Materials and Simulation of Their Cooling Storage

**DOI:** 10.3390/molecules26010107

**Published:** 2020-12-29

**Authors:** Evelyn Reyes-Cueva, Juan Francisco Nicolalde, Javier Martínez-Gómez

**Affiliations:** 1Facultad de Ingeniería y Ciencias Aplicadas, Universidad Internacional SEK, Albert Einstein s/n and 5th, Quito 170302, Ecuador; epreyes.mee@uisek.edu.ec (E.R.-C.); javier.martinez@uisek.edu.ec (J.M.-G.); 2Instituto de Investigación Geológico y Energético (IIGE), Quito 170518, Ecuador

**Keywords:** phase change material, cold thermal energy storage, avocado oil, thermal simulation, material characterization, fatty acids

## Abstract

Environmental problems have been associated with energy consumption and waste management. A solution is the development of renewable materials such as organic phase change materials. Characterization of new materials allows knowing their applications and simulations provide an idea of how they can developed. Consequently, this research is focused on the thermal and chemical characterization of five different avocado seed oils depending on the maturity stage of the seed: 100% unripe, 25% mature-75% unripe, 50% mature-50% unripe, 75% mature-25% unripe, and 100% mature. The characterization was performed by differential scanning calorimetry, Fourier transform infrared spectroscopy, and thermogravimetric analysis. The best oil for natural environments corresponded to 100% matured seed with an enthalpy of fusion of 52.93 $J\cdot g^{-1}$, and a degradation temperature between 241–545 °C. In addition, the FTIR analysis shows that unripe seed oil seems to contain more lipids than a mature one. Furthermore, a simulation with an isothermal box was conducted with the characterized oil with an initial temperature of −14 °C for the isothermal box, −27 °C for the PCM box, and an ambient temperature of 25 °C. The results show that without the PCM the temperature can reach −8 °C and with it is −12 °C after 7 h, proving its application as a cold thermal energy system.

## 1. Introduction

Part of the world’s development depends on technological advances in energy, but economic growth has brought an increase in energetic consumption where fossil fuels represent about 81% of the total. The ${\mathrm{CO}}_{2}$ emitted by the burning of fossil fuels has been identified as a main environmental polluter and a global threat for its contribution to global warming. Sustainable and renewable energy resources aim for the decarbonization of the energy sector, conservation of energy and reduction of emissions [1,2,3,4]. Also, an important part of solving the environmental waste problem is managing agricultural waste. In this sense, optimization of processes like isolation of biophenols from olive leaf waste leads to a 61% ecological reduction and a 49% mitigation of the carbon footprint [5].

Thermal energy storage (TES) makes efficient use of fuels and renewable sources. The technology is feasible as sensible thermal (heat) storage (SHS), latent thermal (heat) storage (LHS), and thermochemical storage (TCS) [6]. Phase change materials (PCMs) are materials that use LHS. The main feature of these materials is the storage or release of great amounts of energy to maintain their temperature when a phase change takes place [7]. It must be considered that the materials that are commonly used in latent heat applications are costly and produced from non-renewable materials, mainly derived from the petrochemical industry, but these types of compounds can also be made from cheap, renewable, raw materials, such as animal fats and vegetable oils [8].

According to the chemical composition of the PCMs, they are mainly classified as organic, inorganic and eutectic mixtures. Organic PCMs are materials with a straight *n*-alkane chain and fatty acids that comprise straight chain hydrocarbons. On the other hand, inorganic PCMs are often metallic materials and salts hydrates capable of maintaining the heat of fusion during long cycling periods. As for the last kind, eutectic mixtures are compounds that have two or more components that change phase together without separating [9,10,11,12]. Due to the big variety of PCMs, their selection aims for specific characteristics. For example, in greenhouses, the objective of the material is to save energy where the best options are salt hydrates, paraffins, and polyethylene glycols [13].

An effective way to contribute to the solution of energy efficiency issues is the use of solid-liquid PCMs, an almost isothermal process [14]. This is the best option to store or maintain the thermal energy of latent heat in small volumes, with minimal loss of heat, and it can achieve excellent thermal comfort for cooling or heating spaces [9]. In general, the TES could use different PCMs, depending on factors like storage capacity, design, and others. Also, it is important to mention that phase change materials are limited by their working temperature [15]. In this sense, applications like domestic refrigeration need an optimal working range. If the phase change temperature is too high, the temperature will increase, reducing the food quality. Contrarily, if the PCM on a fresh food compartment has a working temperature below zero, the food will be frozen [16]. Therefore, it is important to match the working temperature of the PCM to its intended application.

It has been found that PCMs with a melting temperature of −3 °C can prevent the freezing of fresh food and they can be used for cooling during the peak hours, releasing the compressor from work, therefore, saving energetic costs [17].

The applications of PCMs can vary from thermal solar energy storage, water heating systems, low enthalpy energy systems, thermoelectric plants, thermal comfort for buildings, electronics protection, and cooling systems [18]. Specifically, storage of food, drinks, pharmaceutical products, and blood derivatives are applications that require cooling storage [19]. Moreover, the use of cooling systems in fibers, textiles, transport containers, pharmaceutical dispensing systems, informatics stores, memory gadgets and even cancer biodetectors has been reported [10].

Another area of study is cold thermal energy storage (CTES). Its importance lies in the fact that the common cooling systems require major energy consumption. For example, in vehicles and buildings, air conditioning systems consume near 20% of the auxiliary energy [20,21]. Moreover, the application of PCMs in TES can be found in working temperatures from −20 °C to 5 °C, and they can be used in domestic refrigerators and commercial applications like refrigerated trucks, food packing, and medical product applications such as transportation of blood and organs that have strict thermal limitations [22]. Also, live seafood is maintained at a temperature of 5 °C and tropical fruits or rabbit semen need a 10 °C environment for preservation [23].

Important advances on CTES have been made in air conditioning. In this area, various kinds of inorganic PCMs are used like ${\mathrm{LiClO}}_{3}\;3H_{2}O$ with a melting point of 8 °C. Among the organic kind, the paraffin $C_{14}H_{32}$ has a melting point of 6 °C and formic fatty acid a melting point of 7.8 °C. On the other hand, commercial companies like TEAP PCMs (Mumbai, India) have developed a compound with working temperatures from −50 °C to 78 °C [24].

The use of energy storage systems based on fatty acids had increased in recent years due to their good thermodynamic and kinetic performance in the storage of latent heat at low temperatures [25]. Fatty acids are organic compounds [26] that are a good option for new PCMs, especially in solar thermal energy storage applications and being animal or and vegetable derivatives, they have good availability [27]. These oils have been found to have adequate fusion temperature, high fusion latent heat, low cost, very low or non-corrosiveness, low volume variation during the phase change, no toxicity nor inflammability and the don’t present subcooling issues [27,28]. Similarly, since their melting point is within the useful temperature range for thermal energy storage, they can be used in commercial applications as a more attractive alternative, instead of salts and paraffins [29]. Refined and virgin coconut oils were characterized by Kahwaji and White [30], who reported a relatively large heat of fusion and thermal stability making them feasible as a PCM for residential greenhouses.

Some fatty acids have been characterized for cooling applications such as n-pentadecapropyl palmitate, isopropyl palmitate, isopropyl stearate, oleic acid, pelargonic, and formic acid. These acids have melting temperatures of around 10 °C and latent heat from 95 to 247 $\mathrm{kJ}\cdot {\mathrm{kg}}^{-1}$ [24]. Moreover, vegetable oils show good properties at freezing temperatures, like the deodorized soybean oil with a melting point of −5 °C and hydrogenated cottonseed oil with a melting temperature of −0.5 °C [31]. Also, synthesized high chain length fatty acids present thermal reliability after 1000 cycles of melting and freezing [32]. In this sense, the chemical and thermal stability of a PCM determine how the material will endure over time and if it’s feasible for energy storage systems [33].

In this sense, it is important to point out that there is not much information about fatty acids PCMs for freezing temperature uses. Since fatty acids present potential for PCM applications, it is important to know their characteristics. According to the research of Bora et al. [34] gas chromatography analysis has found that the avocado seeds have a fatty acids composition of 32.50% saturated fatty acids, 20.71% monounsaturated acids and 46.73% polyunsaturated acids (PUFA). The latter compounds can also be found in soya beans and corn oils, which are used because of their relative low melting and freezing temperatures [35]. This condition could make a avocado seed oil a PCM for CTES. Moreover, it’s important to mention that the firmness of the pulp and color of the avocado depends on the maturity due its high susceptibility to the enzymatic darkening provoked by the presence of polyphenols [36]. In this sense, there has been previous research on avocado pulp oil [37,38]. However, the characterization of the avocado seed oil at different stages of maturity has not been done.

The avocado is a subtropical fruit that has high concentrations of vitamins and unsaturated fats. Its production is over 4 million tons, with American countries (mainly Mexico) being the largest producers. Nevertheless, production in African, European, and Asian countries has grown. The main market applications of the pulp are in sauces and oils [39]. There are a variety of avocados but the most popular and the one used in this research is the Hass type, which has a pulp with a creamy texture, possesses substantial nutrients and comes from the tree species known as *Persea americana* [40,41]. In the Hass avocado species the seeds represent nearly 12% of the weight of the fruit but are considered a residue, waste, or by-product of the food industry [41,42,43,44]. This seed has been found to be a residual biomass with a good calorific value that makes it feasible as an energy source [45]. For these reasons, this research pays particular attention to take advantage of this residue, give it a new use and avoid it from going directly to the end of its useful life. Although there are some uses for the avocado such as a copolymer binder in the coating industry [46], creation of biomass as well as mineral coal and oil through pyrolysis [47], as a writing ink, and taking advantage of some of its medical advantages in diseases like hypercholesterolemia, hypertension, inflammatory conditions and diabetes. In addition, avocado has been proved to have insecticidal, fungicidal and antimicrobial properties. Likewise, the application of virgin avocado oil as biodiesel has been reported [48,49]. It should be noted that these studies have focused on other areas and not on the conservation of heat at low temperatures.

However, a limitation on the extraction of oil of the avocado seed is that the quantity of obtained oil depends on the extraction technique used [50]. In this sense, the technique of pulverization and Soxhlet extraction delivers 40% of the volume of the original mixture [51].

Chen et al. [52] characterized the paraffin/EG composite as A PCM by means of DSC and simulated its application as LTES, allowing them to predict the behavior of the system during the charging process. Ghahramani and Ahmadi [53] research the performance and durability of PCMs in freezing operations by numerical means, determining the cold storage or release time, the average temperature of the PCM and in the cabin, as well as how the thickness of the PCM would affect the discharge time. Furthermore, Macphee et al. [54] simulated the behavior of encapsulated water for different freezing temperatures, flow rate, and capsule shape, showing that the systems have 99% of energetic efficiency and between 78–82% of exergy efficiency. Numerical calculations by computer allow predicting heat and cold performance in PCMs [55]. In this sense, a simulation by finite elements analysis of a material that has been characterized as PCM helps to know its behavior closer to reality and to take another step towards its mass production or reject its use according to the results. For understanding the thermal behavior of a PCM it is important to perform a 3D simulation for not well studied applications [56].

This research is attractive considering that these technologies reduce energy consumption, and even though avocado oil has been used in the abovementioned applications, avocado seed oil has not been studied much as a PCM. Moreover, the oil comes from a waste of a widely produced fruit, and is not toxic for people nor the environment making it eco-friendly.

The present research has the objective of characterizing five avocado seed oil compounds depending on the maturity stage of the seed, these being 100% unripe, 25% mature-75% unripe, 50% mature-50% unripe, 75% mature-25% unripe, and 100% mature. The best of them will be simulated under a real application by the finite elements method, to prove how efficient is the use of this oil compared to a system without it in CTES applications.

## 2. Materials and Methods

The principal used material was the oil extracted from the seeds of Hass type avocado. Five different samples of oil were taken from mature and unripe seeds. The preparation of this samples started by peeling the thin layer that covers the seed, and then, the seeds were stored at 5 °C until the extraction. The process of obtaining the oil started by dehydrating the seed. Then, the seeds were crashed with a mill, and with the use of a Soxhlet extractor with hexane, the oil was obtained. In addition, to eliminate the solvents the solvent was evaporated on an R-100 rotatory evaporator. This purification was done using the Wesson method [38,57,58]. The extracted oil was stored in an environment of −5 °C in dark vessels without light until its analysis. Table 1 shows the nomenclature used for the characterized compounds.

### 2.1. Characterization of the PCM

#### 2.1.1. Fourier Transformed Infrared Spectroscopy (FTIR) Analysis

With this procedure, we were able to determine the functional groups of the oils. The different samples were analyzed on a FT/IR 4200 FTIR instrument (Jasco, Easton, PA, USA) using the attenuated total reflection technique in the medium infrared range of 400–4000 ${\mathrm{cm}}^{-1}$ as Castonera et al., had done before [36]. Taylor and Rohman also used the same technique to record the spectra in the range of 650–4000 ${\mathrm{cm}}^{-1}.$ One or two drops of the sample were placed in a cell with windows made of KBr or NaBr. The spectra were obtained with a resolution of 4 ${\mathrm{cm}}^{-1}$ with near 3500 points and 60 s of integration time [36]. Lastly, the final trace was plotted after the elimination of the background oxygen and water peaks.

#### 2.1.2. Differential Scanning Calorimetry (DSC) Analysis

The thermal characterization of the material allows to determinate the enthalpies of the samples and their phase change temperatures. The analysis was done with a DSC Q2000 system (TA Instruments, New Castle, DE, USA) in a hermetic chamber with nitrogen as purge gas with a flux of 20 $\mathrm{mL}\cdot \min^{-1}$. The weight of the samples was between 5–10 mg added in hermetically sealed aluminum capsules. In order to avoid any thermal history, the samples were calibrated at 90 °C, keeping them isothermally for 10 min. Then, the samples were cooled down to −80 °C with a ramp of 5 $℃\cdot \min^{-1}$. Finally, they were heated with the same ramp up to 90 °C. For the profile and enthalpy of crystallization, the starting and ending temperatures of the phase change were recorded. Through the thermograms obtained for the enthalpies, the ranges of fusion and crystallization of the samples were determined [57].

#### 2.1.3. Thermogravimetric Analysis (TGA) Analysis

The TGA registered the decomposition that the material faced under the increment of temperature by measuring the mass loss under inert and air environments [32,39,59]. Samples were analyzed in a platinum core with a referential mass between 3 mg–7 mg on a TGA50 instrument (Shimadzu, Kyoto, Japan). Changes in the mass value were recorded from environmental temperature up to 1000 °C at a rate of 10 $℃\cdot \min^{-1}$ under an inert atmosphere of nitrogen and another using air, both with a 30 $\mathrm{mL}\cdot \min^{-1}$ flux.

### 2.2. Simulation

To test the properties of the characterized material, a transient thermal simulation was carried out. In particular, a simulation was used to prove the cooling behavior of the PCM for the application of freezing transportation. First the solids were developed in the Solidworks CAD environment. An isothermal box made of closed cell polypropylene (PP) foam, which according to the software CES-EduPack has applications for packaging medical and scientific products, was modeled. Inside the isothermal box was placed another box made of general purpose high density polyethylene (HDPE) that has applications as film for packing and containers, making this the container for the PCM [60]. On the other hand, a reference of thermal conductivity of the avocado oil has been obtained from the research of Balderas-López et al. [61] and the specific heat comes from the DSC experiments. The properties of both boxes and the chosen PCM to be simulated are presented in Table 2.

For this simulation, the contained air was also modeled as a solid with its corresponding properties since we were interested in its heat transfer. Table 3 displays the boundary conditions for the simulations performed with the Solidworks simulation package and Figure 1a shows a projected view of the assembly of the model with the PCM.

The thermal loads applied were convection on the walls with a solid mesh based on curvature for 29 Jacobian points giving 46,707 elements for the model without PCM and 51,581 for the model with PCM. These considerations are shown on Figure 1b for the model without PCM. The simulation time was 7 h that represents a transport application.

## 3. Results and Discussion

### 3.1. FTIR Analysis Results

The FTIR spectra of the five compounds listed in Table 1 are shown in Figure 2, allowing us to compare their key wavelengths.

Although we planned to record a spectral range between 400 to 4000 ${\mathrm{cm}}^{-1}$, the DSC recorded signals between the range of 750 to 3000 ${\mathrm{cm}}^{-1}.$ It can be seen common bands mainly associated to the recognizable lipids for the stretch and intense peaks which confirms that they are the oils [62].

By analyzing the FTIR study results it was possible to identify peaks that represent the presence of lipids which confirms that the extraction of the avocado seed oil was effective. To be specific, according to Rohman et al. [37], the prominent peaks found at 1742 cm^−1^, 1461 cm^−1^, and 1165 cm^−1^ represent the C=O stretching, -CH_2_ bending, and -C-O stretch or -CH_2_ bending, respectively.

Hence, the presence of those peaks in the materials AA-100T, AA-75T-25M, AA-50T-50M, and AA-25T-75M demonstrate the existence of oils in their composition. The abovementioned peaks are not observed in the AA-100M material. This could be due to the maturity level of the avocado seeds, as the prominence of the peaks increases with the proportion of unripe avocado seed oil. In other words, the unripe avocado seed oil presents a major proportion of lipids because this content is the only difference between the AA-100M and the other materials.

In addition, the presence of alkane terminal carbon –CH_3_ bending vibrations is observed at 1438 cm^−1^ [41]. Moreover, all the materials exhibited a small peak at 1520 cm^−1^. This is attributed to the aromatic bonds present in lignin components [45]. The presence of these bonds could be justified because of an selectivity of the oil extraction process is not 100%.

On the other hand, it is possible to identify that in AA100M the peaks at 2250 and 2350 cm^−1^ are more prominent than in the other compounds. This can be explained by the fact that those peaks represent the presence of CH_2_ stretching, and their presence could be related to the existence of longer fatty acid chains in the mature seed oil. In short, the differences between the materials constituted by unripe avocado seed oil and AA-100M (100% mature avocado seed oil) can be justified by the level of maturity of the seeds. Mature seed oil seems to present less lipids than unripe ones.

The study of Ejiofor et al. [40]. concluded that the avocado seed has a composition high in carbohydrates (49.03 ± 0.02 g/100 g), followed by 17.90 ± 0.14 g/100 g of lipids, 15.55 ± 0.36 g/100 g of protein and moisture (15.10 ± 0.14 g/100 g). As a result, it is possible that the presence of amines seen in the FT-IR spectrograms is due to residues of the Soxhlet extraction process in the avocado seed oils. Appendix A presents the details of the analysis of each compound in Table A1, Table A2, Table A3, Table A4 and Table A5, while Figure A1, Figure A2, Figure A3, Figure A4 and Figure A5 displays the individual curves of each sample with the corresponding frequency labels.

Like the study carried out by Castorena-García and Rojas-López [36] with the help of infrared spectrometry, the components of avocado seed oil were identified, highlighting the identification of type I, II and III amides, and the links that are associated with this oil. Thus, it shows the chemical stability of the material in its different compositions by not presenting elements that can cause instability.

### 3.2. DSC Results

The DSC results shows the phase change temperatures and the total enthalpy of the compounds using a sample of 12 mg. Figure 3 shows the DSC of the compound AA-100M, while the curves of the other compounds are displayed in Figure A6, Figure A7, Figure A8 and Figure A9 in Appendix B. Moreover, Figure 4 represents the curve of the specific heat of the compound AA-100 and how it changes depending on the temperature. When comparing the results, there is an exothermic peak near −47 °C for all the samples. Also, for the fusion curve, there are two endothermal peaks in every compound. Table 4 shows the summary of the crystallization curve and fusion curve results of all the samples. In Appendix C, the extended results with all temperatures and transitions are shown in Table A6.

It can be seen in the obtained results that all proposed materials present a similar range of phase change temperatures, considering that in the fusion process, the first peak appears near −25 °C and the last peak is reached around the 14 °C, suggesting that the fusion phase change takes place between −25 °C to 14 °C, which indicates the suitability of the proposed materials for their application in cooling systems [55]. On the other hand, the storage capacities of the studied materials are not significantly different. In other words, the lowest enthalpy of fusion value (41.93 $J\cdot g^{-1}$) is for the compound AA-100 T, 100% of unripe avocado seed oil, and the highest is 53.97 $J\cdot g^{-1}$ for the compound formed by 50% of unripe avocado seed oil and 50% of mature avocado seed oil. Additionally, a very close value is the latent heat of fusion of the material with 100% mature avocado seed oil (52.93 $J\cdot g^{-1}$). However, the samples show supercooling which is a disadvantage.

As shown, the temperature range of phase change and the latent heat of fusion are not decisive parameters for the selection of the PCM with the best performance in cooling systems. Therefore, it is necessary to analyze another parameter such as its availability. In this way, the compound formed by 100% of mature avocado seed oil, AA-100M, is the best option because this product is commonly found in that state of maturity as waste. The use of the mature avocado seed oil allows the direct exploitation of the avocado seed waste material and at the same time to reduce the quantity of residues that otherwise must be disposed of in landfills without any further usage. In this sense, Figure 5 shows the resultant curve of specific heat displayed for the sample AA-100M at the working temperature.

Although there are many advantages of the usage of PCMs based on residual materials, it is important to analyze their thermal stability after a determined number of heating and cooling cycles, which depicts the usage cycles.

As it can be seen, the latent heat of fusion and the temperature of the PCM for the different compound does not present a considerable variation. Moreover, the peaks appear at close temperatures and the samples with more latent heat are AA-50T-50M and AA-100M.

Both points, where solidification and melting occur in the oils are used for their characterization since these are related to their thermal properties. In this sense, DSC analysis is useful for this determination. It is crucial to take into account that this technique has been used for other oils such as soy and cotton, which is useful to graphically visualize the behavior in the phase change [31].

### 3.3. TGA Results

The TGA results shows the thermal stability of the material and allow knowing at what temperature the degradation of the compounds starts. Table 5 shows the results for the samples AA-100M, AA-75T-25, AA-50T-50M, AA-25T-75M and AA-100T. Appendix D shows the results of the rest of the samples in Figure A10, Figure A11, Figure A12, Figure A13 and Figure A14, despite some noise signals.

The TGA determined that the PCMs do not present degradation under low temperatures, and the loss of mass up to 100 °C is less than 2%. Moreover, it was established that the most resistant sample was the AA-100M one, where its degradation starts at 309 °C in an inert atmosphere and 240 °C for the oxidizing (air) environment.

From the thermogravimetric analyses, it can be indicated that the avocado seed oil PCMs are stable at low temperatures, since the onset of decomposition occurs above 100 °C. Consequently, this material is suitable for low temperature processes according to the criteria of Alper and Aydin [32] and Acurio et al. [33].

Figure 5 shows the TGA analysis of two samples, AA-100M and AA-100T, which are made of 100% mature avocado seed oil and 100% unripe avocado seed oil. Those compounds were selected because they were used as the base components for the three mixtures. This figure depicts the thermal stability of the compounds at low temperatures. Their degradation process starts at temperatures higher than 300 °C.

This allows us to conclude that the materials are suitable for use in cooling systems due to their high thermal resistivity. In addition, it can be observed that there is not water content in the two samples—AA-100M and AA-100T—because there is no weight loss until approximately 250 °C, and water evaporation occurs around 100 °C. This finding is different from the study conducted by Dominguez et al. [39] who observed the presence of water content on the avocado seed. This can be explained by the fact that a crucial step during the oil extraction from the avocado seeds was the evaporation of the solvents and thus the possible water content. On the other hand, they found that the degradation process of the seeds started at temperatures between 250 and 350 °C. Nonetheless, this study found that the mature and unripe avocado seed oil is more thermally stable because their degradation started at approximately 350 °C. To sum up, the presented TGA analysis demonstrated the thermal stability of the compounds used in this study for cooling storage systems.

### 3.4. Material Selection

After characterizing the five different samples, it has been determined that all the samples are constituted primarily by fatty acids (oils). Furthermore, their phase change temperature is around (−25 to 14) °C with a highest enthalpy of fusion of approximately 53 $J\cdot g^{-1}$. All the evaluated materials present a degradation process at temperatures higher than 300 °C making them thermally stable. The abovementioned features of these materials make them suitable for use as PCMs for low temperatures. Nonetheless, it is important to mention that for the purpose of using a PCM for thermal energy storage applications, the enthalpy is very important because its value is directly related with the amount of PCM required for a specific purpose. In other words, low enthalpy values mean a high mass of PCM in the system. Although, the AA-100M sample does not have the highest enthalpy of fusion and even less compared to other PCMs from the review of Oró et al. [19], it has the lowest phase change temperature at −25 °C. Hence, its application will be more suitable for cooling systems. Taking into consideration that this material will be required for commercial applications, the AA-100M material would be the easiest to acquire since its 100% mature seed oil, readily available as a waste product. Consequently, the AA-100M sample was selected to be simulated due to its availability, high enthalpy, and lower temperature of phase change. Table 4 displays the properties of this material since this was simulated.

### 3.5. Simulation Results

Figure 6a shows the simulation of the isothermal box for cold storage without the PCM, and in Figure 6b, the simulation with the PCM as energy store material can be seen, both with its respective thermal scale and a time lapse of 7 h. In addition, Table 6 shows a comparison between the two simulations for average, minimum and maximum temperatures for every part of the system.

The simulation shows that the interior of the box remains colder with the PCM but does not present much difference. However, this difference could allow the product to be stored and with assistance of the PCM will be safer for the needed time. However, the research of Oró et al. [16] proved that a vertical refrigerator with an initial temperature of −22 °C, without PCM, reached nearly 15 °C at 7 h and with the PCM the temperature stayed around 0 °C during the same time lapse. In this sense, this difference lays in the size of the chamber, however, each research behaves in the same way.

Also, Figure 7a shows the gradient of the temperature. It can be seen how this is distributed and stable in all the elements. In Figure 7b, the heat flux is displayed, showing how the heat is retained in every element. Table 7 shows the maximum temperature gradients and heat flux with the PCM and without it. This comparison demonstrates how the PCM retains the energy preventing the heat from flowing to the internal air.

Through the conducted computational simulation, a difference was observed between the use or not use of the PCM in a container for storage and transportation of products at low temperatures. Hence, with an environmental temperature of 25 °C and using a PPE container, without the presence of PCM (sample AA-100M), after 7 h, there was difference of 1 °C. However, the heat flux in the storing compartment was 30 W·$m^{-2}$ lesser by using the PCM, meaning that the products to be stored will be better conserved in comparison of a system without the avocado seed oil PCM.

According to the criteria of Chen [52], it was demonstrated that a simulation of heat transfer in the solid-liquid phase change is necessary to investigate the thermal energy storage behavior. The simulation has proven that the vegetable oils that have polyunsaturated fatty acids have a low phase change temperature allowing them to be used as cold thermal energy storage compounds. Thus, simulation software was needed for the prediction of thermal performance. In this sense, it allowed concluding that the measurements carried out achieved a high agreement with the virtual results, giving good results with the use of PCM. Therefore, a new study can be made using sensors to compare the data obtained in the simulation and the data that a study can produce in a real container. Also, further research could be carried out to look up for different thicknesses of the PCM, how this affect the cooling effects, and which proportion is optimal for commercial uses.

Although avocado seed oil from 100% mature seeds does not have a very high enthalpy and does not has a constant specific heat compared to other PCMs used for low-temperature applications, it has given an acceptable result in the simulation. This could lead to new studies with water-based oils to increase its enthalpy and provide more thermal stability. In this way, the studies carried out by Rasta and Suamir [35] where water-based corn and soybean oils obtained 3 or 5 times higher enthalpy of phase change than the original PCM investigated. Nonetheless, their application works at a temperature between −1 °C and 5 °C.

Since the PCM of AA-100M can be used in temperatures between −25 °C to 14 °C and according to the simulations manages to maintain a −13 °C environment for at least 7 h. Its applications have great potential in the transportation of frozen products, pre-cooked food, blood plasma, bovine and horse semen that use PCMs that works in −21 °C, and pig semen and medicines that needs a temperature near −3 °C [23]

## 4. Conclusions

Five different mixtures of avocado oil of two different maturity stages have been characterized. The samples presented different behaviors from each other. This corroborated that the maturity level of the seed influences the thermal characteristics of the oil. The best of them (sample AA-100M) has been found feasible as a PCM in CTES applications.

The FTIR analysis showed that the avocado seed oil mainly present a lipid profile, with similar behavior in the composition of the different samples. Moreover, it has been determined that unripe seed oil has more lipids than the mature oil

From the thermal analysis, it could be observed that the analyzed oil remained in a liquid state at environmental temperatures and begins its phase change (crystallization) around −3.5 °C, and after solidification when the temperature increases, it begins its melting around −23 °C.

Moreover, the TGA analysis concluded that the degradation of the PCM due to the effect of temperature alone, begins around 300 °C. When oxygen is present, the degradation starts around 220 °C. Hence the effect of thermal degradation cannot affect low temperatures applications.

The mature avocado seed oil, despite of having the second best enthalpy (52.93 $J\cdot g^{-1}$) has a higher temperature range for the phase change. Additionally, it was selected as the most suitable PCM for the following advantages over the other PCMs: availability of seeds of this type, and taking sustainability into account, the pulp in this state is eaten, which is not the case with the pulp of the unripe fruit.

The optimal temperature range to work with the AA-100M sample is between −25 °C and 14 °C, which makes its use valid for low temperature applications. However, as it does not have a high latent heat value, it is necessary to use more mass of the PCM for better results.

The simulation by the finite elements method showed the ability of the PCM to reduce the heat flux of the interior of the isothermal box and to maintain an optimal temperature for cold thermal energy store applications, allowing having cold environments for long periods of time and maintaining −13 °C for 7 h.

## Figures and Tables

**Figure 1 molecules-26-00107-f001:**
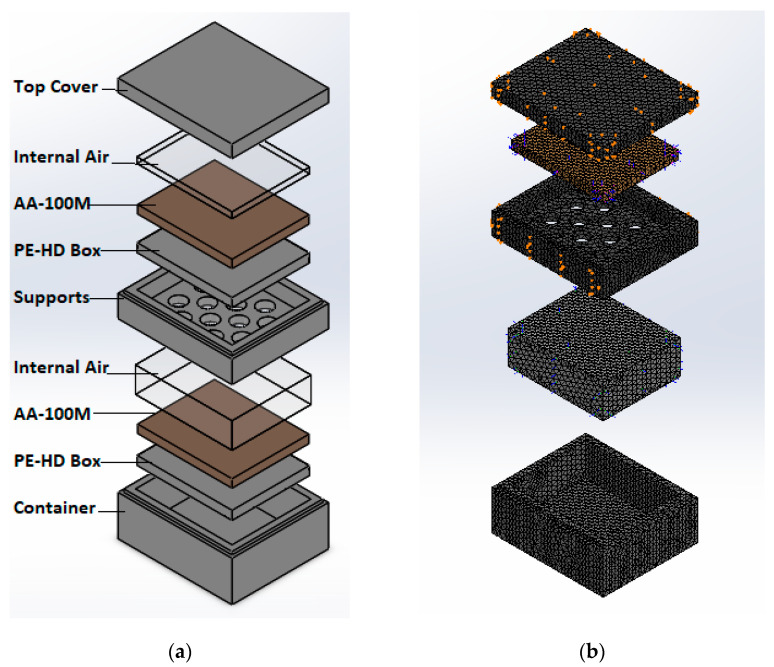
Simulation parameters. (**a**) CAD exploded view of assemble (**b**) Exploded view of the mesh.

**Figure 2 molecules-26-00107-f002:**
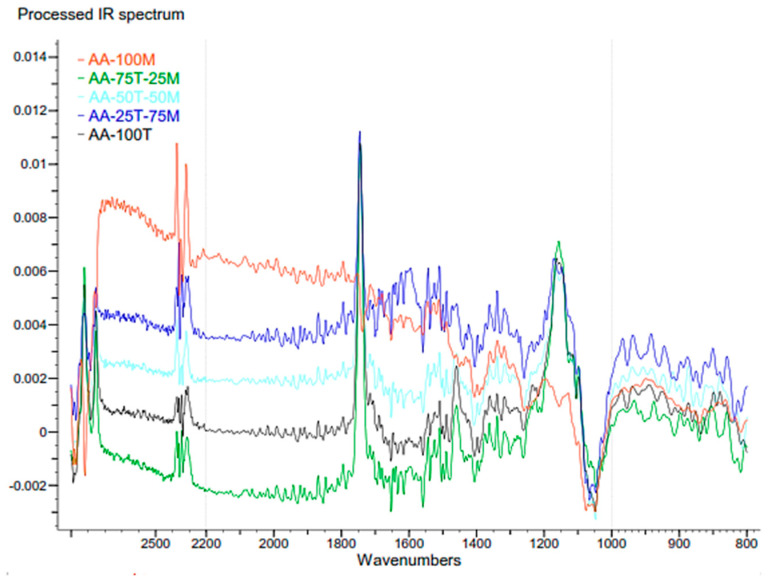
FTIR wave result of the compounds AA-100M; AA-75T-25; AA-50T-50M; AA-25T-75M; AA-100T.

**Figure 3 molecules-26-00107-f003:**
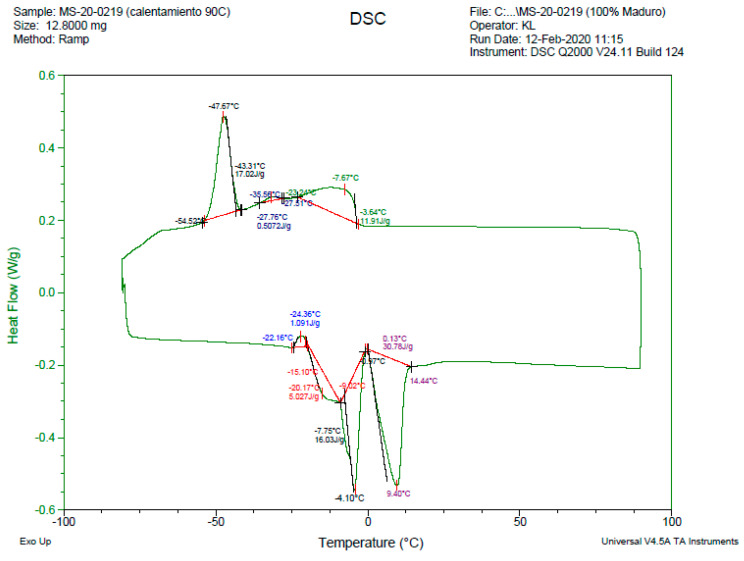
DSC of the sample AA-100M.

**Figure 4 molecules-26-00107-f004:**
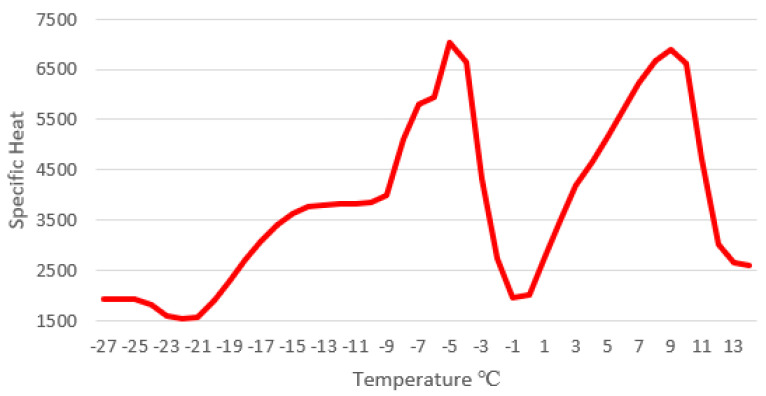
Specific heat temperature for sample AA-100M.

**Figure 5 molecules-26-00107-f005:**
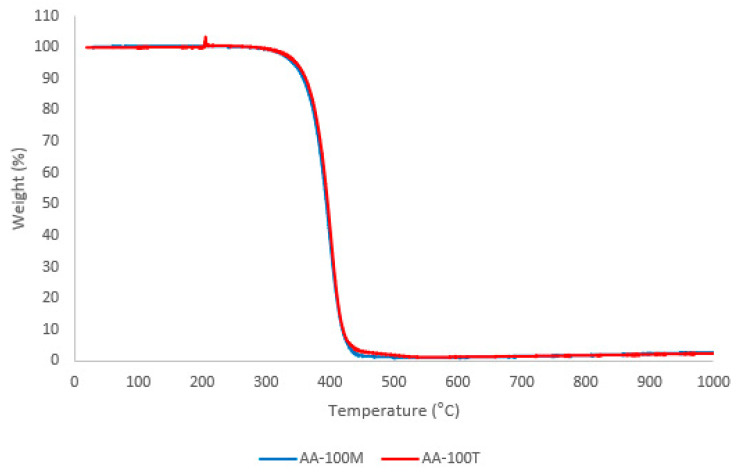
Comparison between TGA curves of samples AA-100M vs. AA-100T.

**Figure 6 molecules-26-00107-f006:**
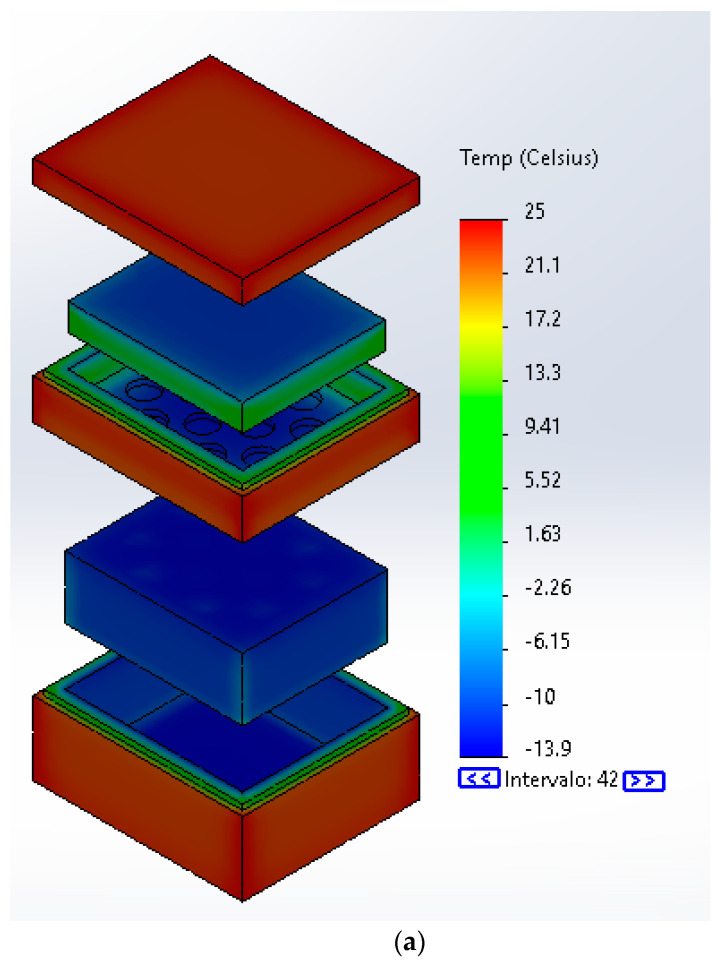
Temperature simulation result. (**a**) Simulation of the isothermal box without PCM (**b**) simulation of the isothermal box with the PCM.

**Figure 7 molecules-26-00107-f007:**
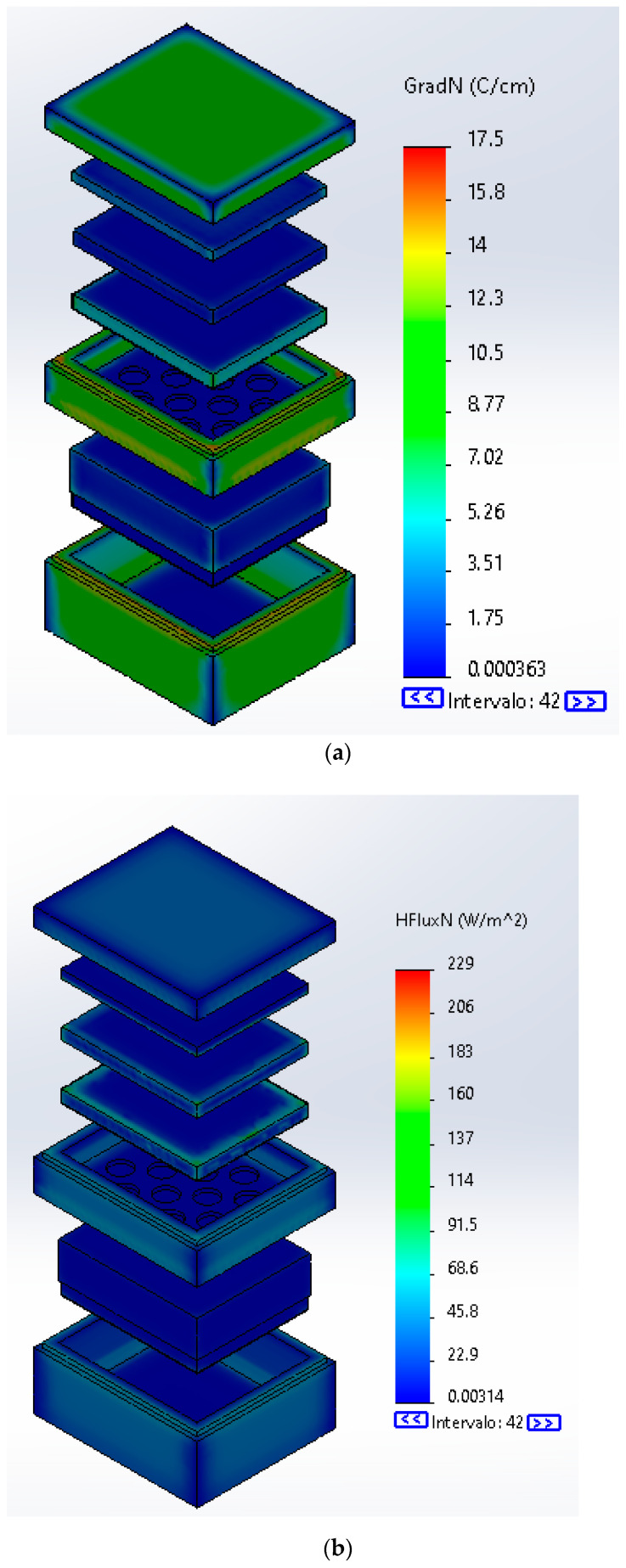
Isothermal box with PCM behavior (**a**) Gradient temperature (**b**) Heat Flux.

**Table 1 molecules-26-00107-t001:** Nomenclature of the combinations of unripe and mature avocado seed oil.

Designation	Unripe Avocado Seed Oil (%)	Mature Avocado Seed Oil (%)
AA-100M	0	100
AA-25T-75M	25	75
AA-50M-50T	50	50
AA-75T-25M	75	25
AA-100T	100	0

**Table 2 molecules-26-00107-t002:** Simulation characteristics for isothermal box, PCM box and PCM.

Material	Density (kg·m^−3^)	Thermal Conductivity (W·m^−1^·°K^−1^)	Specific Heat (J·kg^−1^·°K^−1^)	Size (mm)
Polypropylene (PP)	40	0.040	1870	330 × 270 × 117
High density polyethylene (HDPE)	958	0.48	1779	325 × 265 × 30 × 1
Avocado seed oil (100%M)	687	0.16	1916 (at −27 °C)	-

**Table 3 molecules-26-00107-t003:** Boundary conditions for the thermal simulation.

Solid	Temperature
Isothermal Box	−14 °C
PCM BOX	−27 °C
Environment	25 °C

**Table 4 molecules-26-00107-t004:** DSC results for crystallization and fusion in the five compounds.

Sample	State of Phase	Begin of Transition (°C)	End of Transition (°C)	Peak 1 (°C)	Peak 2 (°C)	Total Enthalpy (J·g^−1^)
AA-100T	Crystallization	−3.8	−54.5	−47.1	−4.2	33.4
	Fusion	−22.2	14.7	−0.6	10.8	41.9
AA-75T-25M	Crystallization	−3.7	−54.5	−47.4	−4.1	32.7
	Fusion	−22.7	14.2	−1.5	10.3	48.4
AA-50T-50M	Crystallization	−3.4	−54.1	−47.9	−4.1	31.9
	Fusion	−23.2	13.1	−2.2	9.6	53.9
AA-25T-75M	Crystallization	−3.6	−54.1	−47.7	−7.7	25.9
	Fusion	−23.9	13.5	−3.1	9.6	46.4
AA-100M	Crystallization	−3.6	−54.5	−47.7	−7.7	29.4
	Fusion	−24.4	14.4	−4.1	9.4	52.9

**Table 5 molecules-26-00107-t005:** TGA results for AA-100M; AA-75T-25; AA-50T-50M; AA-25T-75M; AA-100T.

Sample	Atmosphere	Initial Temp. (°C)	Final Temp (°C)	Initial Mass (mg)	Mass Loss (mg)	Percentage of Loss (%)
AA-100T	Nitrogen	30A	447	6.7	−6.4	96.3
	Air	232	519	5.5	−5.4	98.9
AA-75T-25M	Nitrogen	229	428	3.4	−3.2	93.9
	Air	206	556	3.6	−3.4	95.3
AA-50T-50M	Nitrogen	301	431	3.1	−3.0	96.3
	Air	209	574	3.6	−3.6	98.3
AA-25T-75M	Nitrogen	250	552	5.8	−5.7	97.7
	Air	215	572	5.7	−5.6	97.9
AA-100M	Nitrogen	309	441	4.5	−4.4	96.9
	Air	240	545	5.4	−5.4	98.7

**Table 6 molecules-26-00107-t006:** Temperature simulation comparison.

Material	Initial Temperature (°C)		7 h Average Temperature (°C)		7 h Maximum Temperature (°C)		7 h Minimum Temperature (°C)	
	With PCM	Non PCM	With PCM	Non PCM	With PCM	Non PCM	With PCM	Non PCM
Interior Air	−14	−14	−12.7	−11.8	−6	−1.5	−14	−14
PCM	−27		−13.8		13		−14	
Container	−14	−14	2	2.5	25	25	−11	−14
PCM holder	−14	−14	0	3.6	25	25	−14	−14
Top	−14	−14	10	3.6	25	25	−10	−14

**Table 7 molecules-26-00107-t007:** Maximum thermal gradients and heat flux of the simulation.

Material	Gradient with PCM °C·cm^−1^	Gradient without PCM °C·cm^−1^	Heat Flux with PCM W·m^−2^	Heat Flux without PCM W·m^−2^
Air	16	13	22	52
Container	16	16	66	66
Support	17.5	17.5	141	72
Top	17.5	16	69	65
PCM	17.5	-	69	-

## Data Availability

The data presented in this study are openly available in https://repositorio.uisek.edu.ec/.

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
