# Peer review of "Characterization of Unripe and Mature Avocado Seed Oil in Different Proportions as Phase Change Materials and Simulation of Their Cooling Storage"

_molecules, 2020, doi:10.3390/molecules26010107_

Round 1

Reviewer 1 Report

This paper is in a poor organization and poor quality. The authors shall learn how to write a formal academic paper first. There is no experimental section, so all the results can not be evaluated. The figures are in poor quality. There is no explanation for samples.
English is terrible. There are so many grammar mistakes.

Author Response

Dear Reviewer,

Thank you for the review and the time invested in our paper in order to improve the publication titled “Characterization of the avocado’s young and mature seed oil in different proportions as Phase Change Material and simulation for cooling storage”. It is very important for us to know how we can improve.

Some parts of the article have been re written, to make it easier to understand, all the changes made can be seen with the tracker tool from word. Also, some paragraphs have changed its position to display the information in a more fluid way.

According to your review we have fixed the next points:

  1. This paper is in a poor organization and poor quality.

In this matter we would like to apologize in the sense that the paper had the result part before the method. This organization was written following the journal structure. However, in order to be more understandable we decide to re-write the paper with the method before results, despite the journal’s demands.

  1. The authors shall learn how to write a formal academic paper first.

The paper has been carefully reviewed and re written in a scientific English to match the demands and the seriousness of the publication.

  1. There is no experimental section, so all the results can not be evaluated.

The experimental section is referred as results. In this section the results have been re interpreted and expanded for a more profound analysis. This experimental result can be found as follows:

  • FTIR: From line 226 to 258. Figure 2.
  • DSC: From line 264 to 296. Figure 3. Table 4. Appendix C.
  • TGA: From line 313 to 325. Figure 4. Table 5. Appendix D.

Its important for us to let knowledge that this research aimed for the characterization of 5 different mixtures of avocado seed oils depending on the maturity of the fruit and a simulation of a possible application with the best oil. The experimental application and enhancements to the PCM will be reported in future research as expressed in lines 393-396 and 399.

  1. The figures are in poor quality.

To be more comfortable to see, figures 2, 5 and 6 has been enlarged. Also figure 4 has been developed to provide a better visualization. In the other figure we have done everything that the equipment’s software allows us to and cannot have better quality

  1. There is no explanation for samples.

The explanation for how the samples were obtained and classified is referred in lines 163 to 171 and table 1. With the change of order, where the method precedes the results, despite the journal’s format, this information is more understandable.

  1. English is terrible. There are so many grammar mistakes.

The paper has been reviewed and corrected all the English and grammar mistakes.

If there are any more things that must be changed or recommendations that had been misunderstood, we are pleased to know about them and correct in order to fulfil your expectations for the article to be publish. 

Thank you.

Reviewer 2 Report

The importance of study the Cold Thermal Energy Storage (CTES) is that the common cooling systems have the major energy consumption, for example in vehicles and building the air conditioning systems consumes near the 20% of the auxiliar energy.With a global production exceeding 4 million tons per year in 2011, avocado has become a major agroindustrial commodity. Most of the production and the transformation industry is located in North and Central America, although consumption is growing fast primarily in developed countries like the USA and the European Union. The principal use of the avocado fruit is human consumption, although other applications related to the production of cosmetics, nutritional supplements and livestock feed have been reported. Only the avocado pulp is employed for commercial applications, while other fruit elements like the seed and peel have no practical use and are disposed of by landfilling. Avocado seeds, which represent up to 26 wt % of the fruit mass, are produced in large amounts in centralized avocado transformation plants. Despite their high starch content, the seeds cannot be used for livestock feeding due to the high concentration of polyphenols, which impart a bitter taste and may be toxic at high levels.

A few publications have been dedicated to investigate the energy valorization of avocado seeds into biofuels such as biodiesel and bioethanol . The potential of avocado seeds for biodiesel production is very limited due to the low lipid content of this fraction (<6 %, on a dry matter basis). The production of bioethanol benefits from the high carbohydrate content of the seed (ca. 75 % dmb, mainly in the form of starch, hemicellulose and cellulose ). However, the energy yield of this latter option is limited and its economic viability has not been proven at a commercial scale. Avocado seeds may also be transformed into activated carbon for use in water treatment and gas phase applications . Despite its energy content, direct combustion of avocado seeds for thermal applications has not been reported in the scientific literature.

The subject of the work is interesting and broadly described as you can see. The theoretical introduction does not represent the current state of knowledge. I believe that it should be re-described. In addition, the entire FTIR, TGA and DSC analysis requires the re-presentation of the charts because they are wrong and a new interpretation. In my opinion they are badly described. After these changes, the acceptance of the publication can be misplaced.

Author Response

Dear Reviewer,

Thank you for the review and the time invested in our paper in order to improve the publication titled “Characterization of the avocado’s young and mature seed oil in different proportions as Phase Change Material and simulation for cooling storage”. It is very important for us to know how we can improve.

Some parts of the article have been re written, to make it easier to understand, all the changes made can be seen with the tracker tool from word. Also, some paragraphs have changed its position to display the information in a more fluid way.

According to your review we have fixed the next points:

  1. The theoretical introduction does not represent the current state of knowledge. I believe that it should be re-described. 

The theorical introduction has been expanded and actualized in every introductory topic. Specifically, state of the art of the avocado has been included in lines 114 to 135.

  1. In addition, the entire FTIR, TGA and DSC analysis requires the re-presentation of the charts because they are wrong and a new interpretation. In my opinion they are badly described.

In the results section the FTIR, TGA and DSC results has been re interpreted and expanded for a more profound analysis. This experimental result can be found as follows:

  • FTIR: From line 226 to 258. Figure 2.
  • DSC: From line 264 to 296. Figure 3. Table 4. Appendix C.
  • TGA: From line 313 to 325. Figure 4. Table 5. Appendix D.

If there are any more things that must be changed or recommendations that had been misunderstood, we are pleased to know about them and correct in order to fulfil your expectations for the article to be publish. 

Thank you.

Reviewer 3 Report

The manuscript by Reyes and co-workers discusses the avocado seed oil as a phase change material. The works fits the scope of the journal, the target application is timely and highly important. The work is of interest to a broad audience. However, there are multiple minor and major issues that must be addressed before further consideration by the journal.

1) The results section starts with FTIR and the figure shows the spectra for 5 materials. The abbreviations of these materials are given in the figure itself but they are not identified in the text before. What are these materials?

2) The authors need to add some more critical evaluation of their work including the drawbacks and limitations of the proposed materials and methodologies. The potential impact of the work and next steps should also be clearly communicated.

3) Wavelengths in FTIR (and temperatures) are given down to 2 decimal places which is erroneous. When reporting results, the authors need to be consistent and must take into account the accuracy of the measurement. Reporting 714.41 instead of 714 cm-1 is incorrect, similarly 303,93 should be 304 degC.

4) As prior art, the use of vegetable oils for phase change materials should be mentioned (DOI 10.3390/app9081627). The list of examples in lines 30-39 should also include waste valorization where inefficient PCM is one of the largest energy consuming part (DOI 10.1021/acssuschemeng.9b04245).

5) All the figures are very low resolution. Provide high resolution images.

6) Both the quotient (“x/y”) and negative exponent (“x y-1”) formats are used in the manuscript for units. Either of them should be used consistently, preferably the negative exponent format, which is recommended by the IUPAC.

7) The captions for tables 1 and 2 say ‘curve’ but the table does not show any curve? In general the table and figure captions are short and not informative enough. Provide more descriptive captions for easier understanding of the results.

8) The abstract has 5 abbreviations and only one is spelled out.

9) The keywords should not be merely composed of abbreviations. Avoid using any general abbreviation as keywords and provide meaningful words, specific to the presented research. FTIR, TGA and DSC certainly do not have anything specific and unique for this research.

10) The Materials and Methods section should be before the Results section.

11) The manuscript is full with typos, inconsistent editing, and needs to be thoroughly proofread before any further submission.

Author Response

Dear Reviewer,

Thank you for the review and the time invested in our paper in order to improve the publication titled “Characterization of the avocado’s young and mature seed oil in different proportions as Phase Change Material and simulation for cooling storage”. It is very important for us to know how we can improve.

Some parts of the article have been re written, to make it easier to understand, all the changes made can be seen with the tracker tool from word. Also, some paragraphs have changed its position to display the information in a more fluid way.

According to your review we have fixed the next points:

  1. The results section starts with FTIR and the figure shows the spectra for 5 materials. The abbreviations of these materials are given in the figure itself but they are not identified in the text before. What are these materials?

The explanation for how the samples were obtained and classified is referred in lines 163 to 171 and table 1. With the change of order, where the method precedes the results, despite the journal’s format, this information is more understandable.

  1. The authors need to add some more critical evaluation of their work including the drawbacks and limitations of the proposed materials and methodologies. The potential impact of the work and next steps should also be clearly communicated.

It has been found some limitations on the research that has been expressed in lines 61-65, 107-108, 135-136, 137-139, 397-398. Also, the potential impact is described in lines 403-407 and the next steps in 393-396 and 399.

  1. Wavelengths in FTIR (and temperatures) are given down to 2 decimal places which is erroneous. When reporting results, the authors need to be consistent and must take into account the accuracy of the measurement. Reporting 714.41 instead of 714 cm-1 is incorrect, similarly 303,93 should be 304 degC.

The decimal errors has been corrected all over the paper, nevertheless, in some results that doesn’t have much difference the decimals has been downed to 1 decimal in order to appreciate the difference between results.

  1. As prior art, the use of vegetable oils for phase change materials should be mentioned (DOI 10.3390/app9081627). The list of examples in lines 30-39 should also include waste valorization where inefficient PCM is one of the largest energy consuming part (DOI 10.1021/acssuschemeng.9b04245).

This important references has been added to the theoretical introduction in lines 35-38 and 96-98

  1. All the figures are very low resolution. Provide high resolution images.

To be more comfortable to see, figures 2, 5 and 6 has been enlarged. Also figure 4 has been developed to provide a better visualization. In the other figure we have done everything that the equipment’s software allows us to and cannot have better quality

  1. Both the quotient (“x/y”) and negative exponent (“x y-1”) formats are used in the manuscript for units. Either of them should be used consistently, preferably the negative exponent format, which is recommended by the IUPAC.

The exponent format has been change to the x *y-1 type all over the paper.

  1. The captions for tables 1 and 2 say ‘curve’ but the table does not show any curve? In general the table and figure captions are short and not informative enough. Provide more descriptive captions for easier understanding of the results.

Caption for this tables has been changed and with the reorganization of the paper now you can find them as table 4.

  1. The abstract has 5 abbreviations and only one is spelled out.

All the abbreviations and the abstract has been eliminated

  1. The keywords should not be merely composed of abbreviations. Avoid using any general abbreviation as keywords and provide meaningful words, specific to the presented research. FTIR, TGA and DSC certainly do not have anything specific and unique for this research.

The key words have been changed of more meaningful word and abbreviations eliminated.

  1. The Materials and Methods section should be before the Results section.

In this matter we would like to apologize in the sense that the paper had the result part before the method. This organization was written following the journal structure. However, in order to be more understandable we decide to re-write the paper with the with the method before results, despite the journal’s demands.

  1. The manuscript is full with typos, inconsistent editing, and needs to be thoroughly proofread before any further submission.

The paper has been carefully reviewed and re written in a scientific English to match the demands and the seriousness of the publication. Also, has been reviewed and corrected all the English and grammar mistakes.

If there are any more things that must be changed or recommendations that had been misunderstood, we are pleased to know about them and correct in order to fulfil your expectations for the article to be publish. 

Thank you.

Round 2

Reviewer 1 Report

Although there are many revisions in this manuscript, there are still many problems:
Introduction:
The logic is confusing. I strongly suggest the authors reorganize the content in a logic manner. There shall be a main stream, but now it makes me feel jumping here and there.

Materials and Methods:
You can't obtain a FTIR spectra in a range of 400-4000cm-1 with ATR. It seems that the authors use a liquid cell, not an ATR accessory.
In Table 2, the thermal conductivity of PEHD and avocado seed oil are not correct. In general, their thermal conductivity is ~0.2. Especially, the thermal conductivity of the oil is impossible to be lower than a thermal insulation material such as EPS! Based on wrong basic data, the simulation result is not reliable.

Results and Discussion:
Figure 2 is in poor quality. Obviously, the sample content is too small to give satisfactory S/N ratio. It shall be re-collected.
In Figure 3, not only fusion process, but also crystallization process show two peaks. But in Table 4, only data of one peak is reported.
The authors said that the phase change temperature is -25~14°C, but no data support this.
I am not familiar with simulation, so I will not comment on this part.

Appendix:
In Table A 1, Intensity depends on sample mass, so is not needed. Table A2~Table 5 are not necessary. The corresponding frequency shall be labeled in Figure 2.
All DSC curves show big supercooling. This is a big disadvantage for applications.
In Figures D, there are often noise signals, showing that the TGA instrument is not in a proper status.

There are still many grammar mistakes.

Author Response

Dear Reviewer,

Thank you once more for the review and the time invested in our paper, we value all your work and the opportunity to make our paper “Characterization of the avocado’s unripe and mature seed oil in different proportions as Phase Change Material and simulation for cooling storage” better. The analysis provided allowed us to improve.

According to your review we have fixed the next points:

  1. The logic is confusing. I strongly suggest the authors reorganize the content in a logic manner. There shall be a main stream, but now it makes me feel jumping here and there.

We have edited some of the text in an attempt to make it more logic to follow.

  1. You can't obtain a FTIR spectra in a range of 400-4000cm-1 with ATR. It seems that the authors use a liquid cell, not an ATR accessory.

The reference about this range is explained in lines 179-180 and 233-234.

  1. In Table 2, the thermal conductivity of PEHD and avocado seed oil are not correct. In general, their thermal conductivity is ~0.2. Especially, the thermal conductivity of the oil is impossible to be lower than a thermal insulation material such as EPS! Based on wrong basic data, the simulation result is not reliable.

The data of the materials has been reviewed and found some mistakes that has been corrected. Lines 206-211 supports the characteristics of the materials. Table 2 has been corrected and figure 4 on line 318 has been added to enlarge the thermal characteristics found. With this changes your observation which was right now is corrected. Also, now that the properties of the materials have changed, the results of the simulation has changed as well.

  1. Figure 2 is in poor quality. Obviously, the sample content is too small to give satisfactory S/N ratio. It shall be re-collected.

Figure 2 has the maximum resolution we could record with our equipment, however, figures A1, A2, A3, A4 and A5 has been added to appendix A, which allows to have a better look of the results

  1. In Figure 3, not only fusion process, but also crystallization process show two peaks. But in Table 4, only data of one peak is reported.

This mistake in table 4 has been corrected

  1. The authors said that the phase change temperature is -25~14°C, but no data support this.

The discussion of this result is better explained in lines 281-284

  1. In Table A 1, Intensity depends on sample mass, so is not needed. Table A2~Table 5 are not necessary

We understood that the intensity column is not necessary in every table of appendix A, for this reason we have eliminate that information. Please, let us know if we misunderstood your observation.

  1. The corresponding frequency shall be labeled in Figure 2.

As explained before, the data recorded is the best we could achieve, however, figures in appendix A are labeled with the corresponding frequency.

  1. All DSC curves show big supercooling. This is a big disadvantage for applications.

This observation has been added in line290.

  1. In Figures D, there are often noise signals, showing that the TGA instrument is not in a proper status.

This observation has been added in line 326

  1. There are still many grammar mistakes.

The grammar has been reviewed and corrected in some parts of the paper.

One more, let us thank you for your time and the important observations delivered.

Please, let us know if there are more recommendations or if there was any misunderstanding, we are pleased fulfil all your expectative and correct any mistake for the article to be publish. 

Thank you.

Reviewer 2 Report

The authors referred to all the earlier comments, therefore the work now deserves publication.

Author Response

Dear Reviewer,

Thank you once more for the review and the time invested in our paper, we value all your work and the opportunity to make our paper “Characterization of the avocado’s unripe and mature seed oil in different proportions as Phase Change Material and simulation for cooling storage” better.

Even though we have corrected all your observations and fulfill your expectative, there has been done some changes requested by other reviewer.

Please, if there is any other recommendation let us know.

Thank you.

Reviewer 3 Report

The authors have done a thorough revision and worked through the comments. The figure panels need to be arranged during the typesetting, but the content of the manuscript is acceptable.

Author Response

(The authors gave the same response as above.)
